# Using a data-driven approach for the development and evaluation of phenotype algorithms for systemic lupus erythematosus

**Joel N. Swerdel**[1,2]*, **Darmendra Ramcharran**[1¤], **Jill Hardin**[1,2]

**1** Janssen Research and Development Epidemiology, Titusville, New Jersey, United States of America, **2** Observational Health Data Sciences and Informatics (OHDSI), New York, New York, United States of America

¤ Current address: Safety and Quantitative Innovation, GSK, Waltham, Massachusetts, United States of America

* jswerdel@its.jnj.com

## Abstract

### Background

Systemic lupus erythematosus (SLE) is a chronic autoimmune disease of unknown origin. The objective of this research was to develop phenotype algorithms for SLE suitable for use in epidemiological studies using empirical evidence from observational databases.

### Methods

We used a process for empirically determining and evaluating phenotype algorithms for health conditions to be analyzed in observational research. The process started with a literature search to discover prior algorithms used for SLE. We then used a set of Observational Health Data Sciences and Informatics (OHDSI) open-source tools to refine and validate the algorithms. These included tools to discover codes for SLE that may have been missed in prior studies and to determine possible low specificity and index date misclassification in algorithms for correction.

### Results

We developed four algorithms using our process: two algorithms for prevalent SLE and two for incident SLE. The algorithms for both incident and prevalent cases are comprised of a more specific version and a more sensitive version. Each of the algorithms corrects for possible index date misclassification. After validation, we found the highest positive predictive value estimate for the prevalent, specific algorithm (89%). The highest sensitivity estimate was found for the sensitive, prevalent algorithm (77%).

### Conclusion

We developed phenotype algorithms for SLE using a data-driven approach. The four final algorithms may be used directly in observational studies. The validation of these algorithms

**Data Availability Statement:** The data that support the findings of this study are available from IBM, Optum, JMDC, and IQVIA but restrictions apply to the availability of these data, which were used

under license for the current study, and so are not publicly available. Data are, however, available from the authors upon reasonable request and with permission of IBM, Optum, JMDC, and IQVIA. The authors of the present study had no special privileges in accessing these datasets which other interested researchers would not have. To request access to the datasets used in this study, researchers should use the information, database name and version number, supplied in Table 2.

**Funding:** No sources of funding were used to conduct this study or prepare this manuscript. Johnson and Johnson will be the sponsor of Open Access, if applicable. The funders had no role in study design, data collection and analysis, decision to publish, or preparation of the manuscript.

**Competing interests:** Authors JS and JH are employees of Janssen Research and Development and shareholders of Johnson & Johnson. DR was an employee of Janssen Research and Development and shareholder of Johnson & Johnson and is currently affiliated with GSK.

**Abbreviations:** ANA, Anti-nuclear antibody; CCAE, Commercial Claims and Encounters; DMARD, Disease-modifying antirheumatic drugs; EHR, Electronic health records; ICD-9, International Classification of Diseases, Ninth Revision; MDCD, Medicaid; MDCR, Medicare; OHDSI, Observational Health Data Sciences and Informatics; PPV, Positive predictive value; PY, Person years; QBA, Quantitative bias analysis; RWE, Real-world evidence; SLE, Systemic lupus erythematosus; SMD, Standard mean difference; SNOMED, Systemized Nomenclature of Medicine; YO, Year old.

provides researchers an added measure of confidence that the algorithms are selecting subjects correctly and allows for the application of quantitative bias analysis.

## Introduction

Systemic lupus erythematosus (SLE) is a chronic autoimmune disease of unknown origin. Clinical manifestations include fatigue, arthropathy, and involvement of nearly all organ systems, particularly cardiac and renal [1–4]. A review by Stojan and Petri of research on multi-country incidence rate estimates found the incidence rate of SLE to be between 1–9 cases per 100,000 person-years (PY) [5].

The use of real-world evidence (RWE) from observational data, including administrative claims and electronic health records (EHR) datasets, is critical for studying the epidemiology and clinical manifestations of SLE. Developing and applying accurate phenotype algorithms is central for using observational data for analyses of health conditions. We performed a literature search and found 58 journal articles which included phenotype algorithms for SLE. A phenotype algorithm is the translation of the case definition of a health condition or phenotype into an executable algorithm based on clinical data elements in a database [6]. The algorithms used included those with 1 or more codes for SLE in a patient's record. Many included requirements for laboratory results from anti-nuclear antibody (ANA) tests. Others included prescriptions for anti-malarial drugs or oral corticosteroids. We found five studies validating SLE algorithms using clinical adjudication after 2010. Each study was performed on a single dataset. The research by Barnado et al provided the most recent example of the validation of algorithms for SLE. They evaluated a wide variety of algorithms using variations of the number of codes in a patient's record for SLE (1–4 codes), presence of drugs for SLE including anti-malarials, corticosteroids, and disease-modifying antirheumatic drugs (DMARD), and results from ANA tests. They performed their analyses in a single database, the Vanderbilt University Medical Center's Electronic Health Record system. The algorithms were evaluated in a group of 200 randomly selected patients whose medical charts were reviewed and adjudicated by clinicians to confirm or refute the presence of SLE. The best performing algorithm was one with $\geq$ 3 SLE International Classification of Diseases, Ninth Revision (ICD-9) code 710.0 counts, ANA positive laboratory result, ever use of DMARDs, and ever corticosteroids use. They found a positive predictive value (PPV) of 91%. They also found a PPV of 95% when they excluded subjects with Dermatomyositis (ICD-9 code 710.3) and Systemic Sclerosis (ICD-9 code 710.1). In another algorithm with $\geq$ 3 SLE ICD-9 code counts and ever antimalarial use, they found a PPV of 88% without excluding subjects with Dermatomyositis and Systemic Sclerosis and a PPV of 91% when excluding subjects with Dermatomyositis and Systemic Sclerosis. Limitations of this study included the use of data from a single health center and the calculation of sensitivity based on subjects with a mention of SLE in their record. Specificity was not measured in this study.

The objective of this research was to develop and evaluate phenotype algorithms for SLE suitable for use in epidemiological studies using empirical evidence from data in observational databases. The development process was performed on multiple databases from several countries to increase generalizability. The evaluation process included all performance characteristics, including sensitivity and specificity, and were assessed independent of the inclusion of SLE codes in each subject's health record and without utilizing medical chart review.

## Materials and methods

We applied a rigorous process to develop the algorithms in this study. We used an extensive literature review (details and results are in S1 File), along with several tools within the Observational Health Data Sciences and Informatics (OHDSI) tool stack for empirical analysis. The goal of instituting this process was to enhance the science of phenotype algorithm development. The full details of the process are described in S2 File. We developed and evaluated four phenotype algorithms for SLE to be analyzed in observational research (Table 1): two incident algorithms requiring at least 365 days prior look-back and at least one SLE code ("Incident, 1X") or at least two codes with a code for SLE 31–365 days after the first code ("Incident, 2X"); two prevalent algorithms requiring at least one SLE code ("Prevalent, 1X") or at least two codes with a code for SLE 31–365 days after the first code ("Prevalent, 2X"). Requiring a 365 day look-back for the incident algorithm will impact the sensitivity of the algorithm especially in databases where subjects are observed for shorter periods of time. Each of the algorithms was corrected for possible cohort entry date (index date) misclassification. This was achieved by allowing for signs and symptoms for SLE to mark the start of the condition if these were followed by a code for SLE within 90 days of the signs or symptoms (Fig 1). We found 90 days to be optimal to achieve good coverage of prior signs, symptoms, and drugs. In the period 365 to 91 days prior to the first diagnosis code for SLE, the rates of these signs and symptoms were significantly lower than in the 90 to 0 days prior to index date. Thus the use of 90 days prior to index as the optimal time period choice. Signs and symptoms included codes for malaise, joint pain, and low back pain, consistent with the signs and symptoms of SLE [2,7–9]. There were also prescriptions for drugs often used in the treatment of SLE and its symptoms, such as prednisone and methylprednisolone. The presence of these diagnosis and drug codes in the 90 days prior to the index date may indicate that the condition began prior to the appearance of the first SLE diagnosis code and the clinical diagnosis. From these results we inferred that there was likely index date misclassification and corrections to account for the prior signs, symptoms, and drugs were applied to improve the index date accuracy. The complete set of ICD, Read, and SNOMED codes used in the algorithms are in the S3 File.

We evaluated the algorithms against a network of nine observational databases. All databases were standardized to the OMOP Common Data Model, which provides the capability for the phenotype algorithms to be developed and consistently applied across the data sources. The nine databases were a mix of administrative insurance claims, electronic health records, and general practitioner databases from Germany, France, Australia, Japan, and US. The description and details for each of the databases are in Table 2. For each of the cohorts in each of the databases, we extracted patient-level data on patient demographics, diagnostic conditions, laboratory measurements, diagnostic and medical procedures, and drug exposures. The

**Table 1. Names, characteristics, and external links of phenotypes for Systemic lupus erythematosus (SLE).**

| Cohort | Attributes | Github Link | Cohort Diagnostics Shiny ID |
|---|---|---|---|
| Systemic lupus erythematosus incident and correction for index date | incident population, sensitive performance characteristics | Cohort 1 | C2–2409 |
| Systemic lupus erythematosus prevalent and correction for index date | prevalent population, sensitive performance characteristics | Cohort 2 | C5–3627 |
| Systemic lupus erythematosus incident with 2nd diagnosis code and correction for index date | incident population, specific performance characteristics | Cohort 3 | C3–2410 |
| Systemic lupus erythematosus prevalent with 2nd diagnosis code and correction for index date | prevalent population, specific performance characteristics | Cohort 4 | C6–3628 |

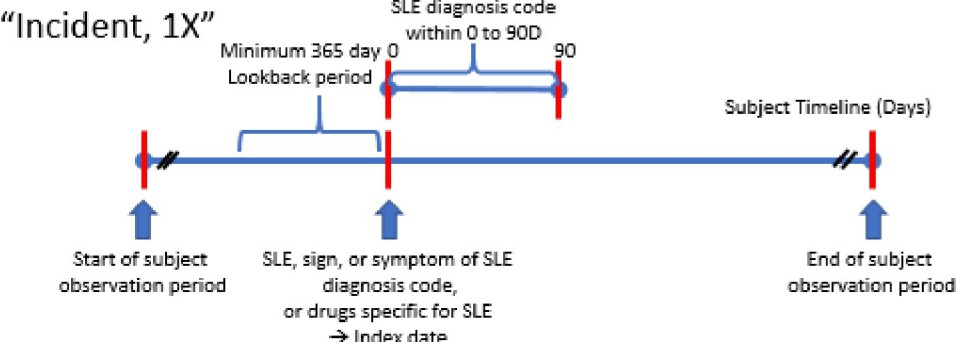

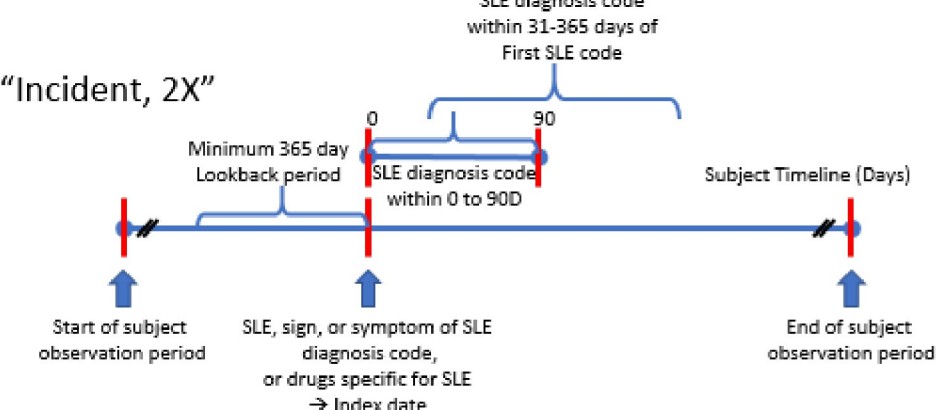

**Fig 1. Diagram of the timeline for the phenotype algorithms for Systemic lupus erythematosus.**

Optum and IBM MarketScan databases used in this study were reviewed by the New England Institutional Review Board (IRB) and were determined to be exempt from broad IRB approval, as this research project did not involve human subject research. Based on Ethical Guidelines for Epidemiological Research issued by the Japanese Ministry of Health, Labor and Welfare, ethics approval and informed consent for the JMDC database were not applicable for this study. The data from IQVIA were deemed commercial assets and there was no IRB applicable to the usage and dissemination of these result sets or required registration of the protocol with additional ethics oversight.

We applied the OHDSI CohortDiagnostics tool (https://github.com/OHDSI/CohortDiagnostics) to evaluate and compare algorithms at a population-level, characterizing overall count, incidence over time, index date breakdown, cohort overlap, and temporal characterization.

We applied the PheValuator [10] method to evaluate the performance characteristics of the algorithms against the databases. This method provides the complete set of performance

**Table 2. Description of databases used in the study.**

| Name (Abbreviation) | Years | Country | Data Type | Clinical Visits included | Number of Persons (millions) | Average Age at First Observation | Percent Female | Median Length of Follow-up (years) |
|---|---|---|---|---|---|---|---|---|
| IQVIA Australian Longitudinal Patient Data v1945 (Australia) | 1996–2020 | Australia | General practitioner data | Outpatient | 5 | 37 | 22* | 0.5 |
| IBM MarketScan Commercial Claims and Encounters v2136 (CCAE) | 2000–2021 | US | Insurance Claims | Inpatient/ outpatient | 157 | 31 | 51 | 1.56 |
| IQVIA Disease Analyzer–France v1943 (France) | 2016–2021 | France | General practitioner data | Outpatient | 4 | 37 | 52 | 0.9 |
| IQVIA Disease Analyzer–Germany v1944 (Germany) | 2011–2021 | Germany | General practitioner data with supplemental data from participating specialists | Outpatient | 31 | 43 | 56 | 0.5 |
| Japan Medical Data Center v2129 (JMDC) | 2000–2021 | Japan | Insurance Claims | Inpatient/ outpatient | 12 | 31 | 49 | 3.29 |
| IBM MarketScan Multi-State Medicaid v2128 (MDCD) | 2006–2020 | US | Insurance Claims | Inpatient/ outpatient | 31 | 23 | 56 | 1.52 |
| IBM MarketScan Medicare Supplemental v2135 (MDCR) | 2000–2021 | US | Insurance Claims | Inpatient/ outpatient | 10 | 71 | 55 | 2.46 |
| Optum Clinformatics Extended Data Mart—Date of Death v2050 (Optum) | 2007–2021 | US | Insurance Claims | Inpatient/ outpatient | 71 | 37 | 51 | 1.48 |
| Optum Pan-Therapeutic Electronic Health Records v2137 (Optum EHR) | 2007–2021 | US | Electronic health records | Inpatient/ outpatient | 99 | 37 | 53 | 2.63 |

* 59% of subjects do not have a designated sex.

characteristics, i.e., sensitivity, specificity, and positive and negative predictive value. Using this method, we also evaluated algorithms from the Barnado et al study for comparison [11]. The two algorithms from Barnado et al were "Systemic Lupus 3X plus ever anti-malarial drugs" and "Systemic Lupus 3X plus ever anti-malarial drugs excluding dermatomyositis and systemic sclerosis". Using the semi-automated phenotype algorithm evaluation method PheValuator, we eliminated the need for obtaining and reviewing subject's records. While algorithm validation results from chart review are considered the "gold standard", we have compared the results from PheValuator with prior studies using chart review and found excellent agreement between the two methods [12].

The demographic and clinical characteristics of those determined by the Incident, 2X algorithm were compared to those from a non-SLE randomly selected cohort of subjects. Non-SLE subjects were matched to the SLE cohort by age, sex, and year and month of SLE diagnosis in a ratio of 1 SLE:10 non-SLE subjects. The year and month of SLE diagnosis with the first clinical visit was matched in the same year and month in the randomly selected subject. We also required the same minimum look-back period, 365 days, in the matched cohort as was specified in the SLE cohort. Comparisons were made for characteristics observed in the year prior to SLE diagnosis (SLE cohort) or the matching visit date (non-SLE cohort). Standardized difference of the mean were calculated to assess difference between the cohorts [13].

The source code to implement the process used to develop and evaluate the algorithms is publicly available at (https://github.com/OHDSI/PhenotypeEvaluations/tree/main/SLE). This repository contains the literature search algorithm, the cohort definitions in JSON format, and the code to run CohortDiagnostics and PheValuator.

**Table 3. Comparison of subject counts across databases for the selected algorithms for SLE.**

| Database | Systemic lupus erythematosus incident and correction for index date | Systemic lupus erythematosus incident with 2nd dx and correction for index date | Systemic lupus erythematosus prevalent and correction for index date | Systemic lupus erythematosus prevalent with 2nd dx and correction for index date |
|---|---|---|---|---|
| Australia | 319 | 67 | 951 | 252 |
| CCAE | 183086 | 60225 | 419196 | 196490 |
| France | 117 | 26 | 260 | 75 |
| Germany | 3687 | 710 | 10137 | 1887 |
| JMDC | 10404 | 5528 | 23921 | 17063 |
| MDCD | 35547 | 15152 | 88097 | 47474 |
| MDCR | 24680 | 8039 | 50183 | 22496 |
| Optum | 101025 | 32891 | 234861 | 108853 |
| Optum EHR | 142159 | 59031 | 217048 | 88152 |

Australia—IQVIA Australian Longitudinal Patient Data; CCAE—IBM® MarketScan® Commercial Database; Optum—Optum® Clinformatics® Data Mart; France—IQVIA Disease Analyzer–France; Germany—IQVIA Disease Analyzer–Germany; JMDC—Japan Medical Data Center; MDCD IBM® MarketScan® Multi-State Medicaid Database; MDCR—IBM® MarketScan® Medicare Supplemental Database; Optum EHR—Optum® longitudinal EHR repository.

## Results

We examined the characteristics of the subjects included by the algorithms. These characteristics may be viewed interactively at https://data.ohdsi.org/SLECohortDiagnostics. Subject counts in each of the databases ranged from about 117 subjects in France to about 180K in CCAE for the Incident, 1X algorithm (Table 3). The counts were as expected based on the relative sizes of the databases indicating that all codes used were appropriate for each database. The counts were much higher in the US databases compared to the databases from outside the US. The reduction in the number of subjects in the Incident, 1X algorithm compared to the Incident, 2X algorithm was significant, ranging from about an 80% reduction in Germany and Australia to about a 50% reduction in Japan. This is graphically depicted in Fig 2 which shows the proportion of overlap between the algorithms in each database. Fig 2 also shows that the Incident, 2X algorithm is a proper subset of Incident, 1X algorithm, i.e., all subjects in the 2X algorithm were also in the 1X algorithm.

We compared the diagnosed conditions, prescribed drugs, laboratory measurements, and clinical procedures of the subjects included in cohorts defined by the algorithms to see how

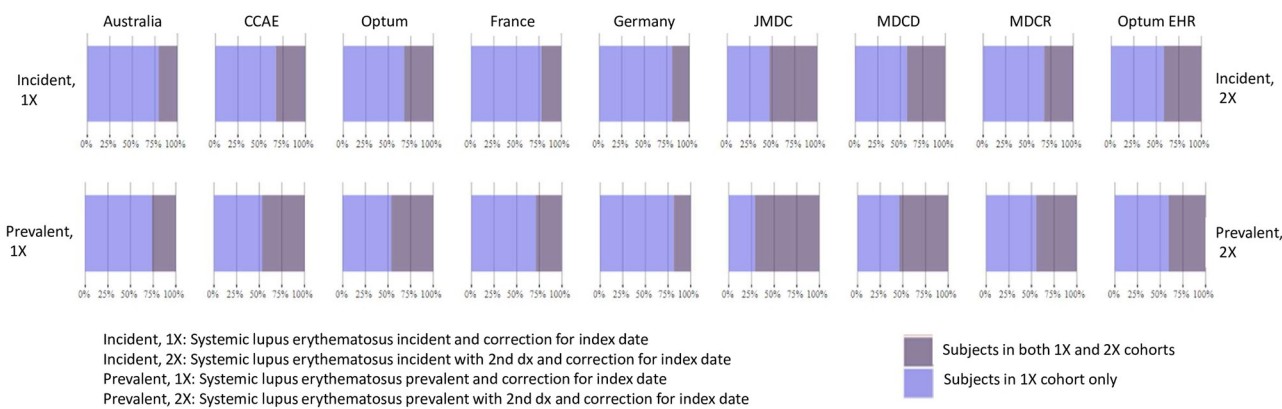

Incident, 1X: Systemic lupus erythematosus incident and correction for index date
Incident, 2X: Systemic lupus erythematosus incident with 2nd dx and correction for index date
Prevalent, 1X: Systemic lupus erythematosus prevalent and correction for index date
Prevalent, 2X: Systemic lupus erythematosus prevalent with 2nd dx and correction for index date

Subjects in both 1X and 2X cohorts
Subjects in 1X cohort only

**Fig 2. Graphical depiction of the overlap in subjects between the two incidence cohorts and the two prevalence cohorts.**

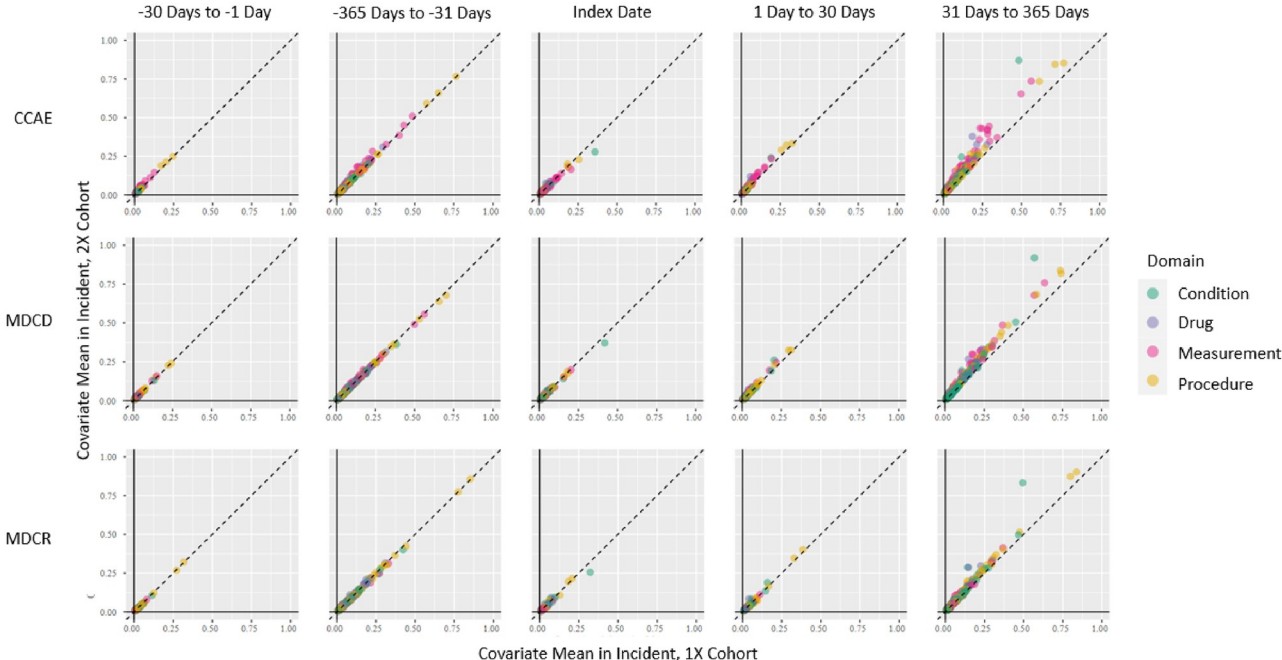

**Fig 3. Comparison between proportion of subjects in the "Incident, 1X algorithm" compared to the "Incident, 2X algorithm" for three selected datasets with different demographic characteristics.** Points closest to the diagonal indicate similar proportions between the comparators; points farther from the diagonal indicate more disparate proportions.

the populations differed. Fig 3 shows a comparison between subjects in the Incident, 1X algorithm compared to the Incident, 2X algorithm for three selected datasets with different demographic characteristics. The CCAE database, an insurance claims database of employed individuals and their families generally under 65 years old, showed the largest disparity between the two algorithms in the period 31–365 days after the index date. Some of the differences were from higher proportions of diagnosis codes for SLE (87% Incident, 2X v. 48% Incident, 1X, standard mean difference (SMD) 0.64) and prescriptions for hydroxychloroquine (38% Incident, 2X v. 19% Incident, 1X, SMD 0.31). There were also differences, albeit fewer, in the Medicaid (MDCD) population, those generally of lower socioeconomic status. This dataset also showed differences in diagnosis codes for SLE (92% Incident, 2X v. 57% Incident, 1X, standard mean difference (SMD) 0.61) and prescriptions for hydroxychloroquine (27% Incident, 2X v. 15% Incident, 1X, SMD 0.21). The Medicare (MDCR) dataset, generally individuals 65 years and older, showed fewer significant differences in proportions between the two cohorts. However, there were also differences in diagnosis codes for SLE (83% Incident, 2X v. 49% Incident, 1X, standard mean difference (SMD) 0.54) and prescriptions for hydroxychloroquine (29% Incident, 2X v. 15% Incident, 1X, SMD 0.24). Overall, the relative proportions between the two cohorts for the majority of the characteristics in MDCR were closer to the 45° line, indicating similar proportions between the cohorts compared to Commercial Claims and Encounters (CCAE) and MDCD.

We examined the Incident, 2X algorithm for subject characteristics across the databases. A higher proportion of females compared to males with SLE were identified. The largest disproportionality was in MDCD where 91% of the subjects were female. Japan had the lowest disproportionality by sex with 68% of the subjects female. The type of clinical visit on first diagnosis of SLE was most commonly an outpatient or office visit. Less than 5% of the first diagnoses were made in an inpatient or emergency room visit with the exception being MDCD where

about 25% of the first diagnoses were made in an emergency room visit. This follows with other published studies examining the higher use of the emergency room in MDCD recipients [14]. Several differences were found in the demographic and clinical characteristics of the Incident, 2X algorithm compared to the matched non-SLE cohort. Differences in the standardized difference of the mean greater than 0.1 are considered imbalanced [15]. Imbalanced characteristics between the two cohorts included Black and White race (in MDCD and Optum EHR), diagnoses for rheumatoid arthritis, heart disease, and renal impairment. There were also differences in prescriptions for immunosuppressants and anti-thrombotic agents.

We also examined the index event, i.e., the diagnosis code that initiated the subject into the cohort. In the Incident, 2X algorithm, the most prevalent index event was a diagnosis code of "Systemic lupus erythematosus" (SNOMED code 257628; ICD-10CM and ICD-10GM M32.9 ("Systemic lupus erythematosus, unspecified"); ICD-9CM 710.0 ("Systemic lupus erythematosus")). There were a significant proportion of subjects whose index event was for a sign or symptom of SLE such as malaise, fatigue, anemia, or low back pain. These accounted for about 40–50% of the index events, indicating that there was likely a large amount of index date misclassification in subjects with SLE.

Incidence rates for SLE were similar from 2015 to 2020 across many of the databases. This may be partially attributed to the standardized use of ICD-10 coding starting in the US in 2015. In CCAE, MDCD, MDCR, Optum EHR, and Japan the rates were about 16 per 100,000 person years (PY). The rates in Germany were 1 per 100,000 PY. Rates in Australia and France varied considerably, likely due to the small sample size. The small number of SLE subjects in the Australian and French databases is likely due to the database being limited to general practitioners. In other databases which include specialists, such as rheumatologists, the sample size and incidence rates are higher and more stable. The incidence rates peaked in subjects in the 50–59 year old (YO) age group. One exception was in Japan where the incidence rates continued to increase with age through ages 70-79YO. Incidence rates in females were about 23–30 per 100,000 PY with the exception of Japan where the rate in females was about 18 per 100,000 PY. The incidence rates in males ranged from about 3–8 per 100,000 PY.

Evaluation of the performance characteristics of the four final algorithms was assessed using the PheValuator method (Table 4). Due to low subject counts, we were unable to calculate the performance characteristics for Australia and France. PheValuator requires a minimum of 200 subjects with a high likelihood of having SLE to produce an accurate model. This number was not satisfied in the Australian or French databases. As noted earlier, this may be due to the limitation of these databases to general practitioners. In general, the highest PPV estimates were for the two algorithms where a second diagnosis code for SLE was required 31–365 days after index. The highest sensitivity estimates were found in the two prevalent cohorts. The mean PPVs for the two algorithms where a second code was required was 87% (incident) and 88% (prevalent). This was reduced to 57% (incident) and 58% (prevalent) where only a single diagnosis code for SLE was required. The sensitivity for the two prevalent algorithms were 82% (single code required) and 55% (two codes required). This decreased to 39% (single code required) and 24% (two codes required) for the two incident cohorts. The highest mean F1 score was found in the prevalent, single code algorithms (65%) and the lowest was in the incident algorithms requiring two SLE diagnosis codes (37%). PPV estimates were higher in the US databases compared to Germany and Japan while the sensitivity estimates were similar between the databases. The estimates for PPV for the Barnado et al algorithms were 90% for both the algorithm using ≥ 3 SLE ICD-9 code counts and ever antimalarial use, either with or without exclusion of subjects with Dermatomyositis and Systemic Sclerosis. This is similar to that found by Barnado et al (88% including subjects with Dermatomyositis and Systemic Sclerosis; 91% excluding subjects with Dermatomyositis and Systemic Sclerosis).

**Table 4. Performance characteristics of the algorithms for SLE.**

| Phenotype Algorithm | Database | Sensitivity (95% CI) | PPV (95% CI) | Specificity (95% CI) | NPV (95% CI) |
|---|---|---|---|---|---|
| Systemic Lupus 3X plus ever anti-malarial drugs (per Barnado, 2017) [11] | CCAE | 0.437 (0.424–0.449) | 0.969 (0.962–0.975) | 1.000 (1.000–1.000) | 0.998 (0.998–0.998) |
| | Optum | 0.425 (0.415–0.435) | 0.939 (0.931–0.946) | 1.000 (1.000–1.000) | 0.997 (0.997–0.997) |
| | Germany | 0.388 (0.327–0.453) | 0.817 (0.735–0.883) | 1.000 (1.000–1.000) | 1.000 (1.000–1.000) |
| | JMDC | 0.069 (0.056–0.085) | 0.706 (0.615–0.786) | 1.000 (1.000–1.000) | 0.999 (0.999–0.999) |
| | MDCD | 0.279 (0.271–0.288) | 0.953 (0.945–0.960) | 1.000 (1.000–1.000) | 0.995 (0.995–0.995) |
| | MDCR | 0.292 (0.283–0.301) | 0.953 (0.945–0.960) | 1.000 (1.000–1.000) | 0.996 (0.996–0.996) |
| | Optum EHR | 0.317 (0.305–0.329) | 0.960 (0.950–0.969) | 1.000 (1.000–1.000) | 0.998 (0.998–0.998) |
| Systemic Lupus 3X plus ever anti-malarial drugs excluding DM and SSc (per Barnado, 2017) [11] | CCAE | 0.408 (0.395–0.420) | 0.969 (0.961–0.975) | 1.000 (1.000–1.000) | 0.998 (0.998–0.998) |
| | Optum | 0.401 (0.391–0.411) | 0.939 (0.931–0.946) | 1.000 (1.000–1.000) | 0.997 (0.997–0.997) |
| | Germany | 0.347 (0.287–0.411) | 0.808 (0.719–0.878) | 1.000 (1.000–1.000) | 1.000 (1.000–1.000) |
| | JMDC | 0.063 (0.050–0.079) | 0.706 (0.612–0.790) | 1.000 (1.000–1.000) | 0.999 (0.999–0.999) |
| | MDCD | 0.261 (0.253–0.269) | 0.953 (0.945–0.960) | 1.000 (1.000–1.000) | 0.995 (0.995–0.995) |
| | MDCR | 0.273 (0.264–0.282) | 0.953 (0.945–0.961) | 1.000 (1.000–1.000) | 0.996 (0.996–0.996) |
| | Optum EHR | 0.305 (0.293–0.317) | 0.960 (0.950–0.969) | 1.000 (1.000–1.000) | 0.998 (0.998–0.998) |
| Systemic lupus erythematosus incident and correction for index date | CCAE | 0.368 (0.356–0.380) | 0.598 (0.582–0.614) | 0.999 (0.999–0.999) | 0.998 (0.998–0.998) |
| | Optum | 0.323 (0.313–0.332) | 0.538 (0.525–0.551) | 0.999 (0.999–0.999) | 0.997 (0.997–0.997) |
| | Germany | 0.368 (0.307–0.432) | 0.335 (0.278–0.395) | 1.000 (1.000–1.000) | 1.000 (1.000–1.000) |
| | JMDC | 0.572 (0.544–0.600) | 0.484 (0.458–0.511) | 1.000 (1.000–1.000) | 1.000 (1.000–1.000) |
| | MDCD | 0.377 (0.369–0.386) | 0.694 (0.682–0.705) | 0.999 (0.999–0.999) | 0.996 (0.996–0.996) |
| | MDCR | 0.314 (0.304–0.323) | 0.545 (0.532–0.558) | 0.999 (0.998–0.999) | 0.996 (0.996–0.996) |
| | Optum EHR | 0.434 (0.421–0.447) | 0.808 (0.793–0.822) | 1.000 (1.000–1.000) | 0.998 (0.998–0.998) |
| Systemic lupus erythematosus prevalent and correction for index date | CCAE | 0.855 (0.846–0.864) | 0.633 (0.622–0.643) | 0.998 (0.998–0.998) | 1.000 (1.000–1.000) |
| | Optum | 0.862 (0.855–0.869) | 0.625 (0.617–0.634) | 0.997 (0.997–0.998) | 0.999 (0.999–0.999) |
| | Germany | 0.983 (0.958–0.995) | 0.244 (0.217–0.272) | 1.000 (1.000–1.000) | 1.000 (1.000–1.000) |
| | JMDC | 0.859 (0.838–0.878) | 0.495 (0.474–0.517) | 0.999 (0.999–0.999) | 1.000 (1.000–1.000) |
| | MDCD | 0.816 (0.809–0.823) | 0.712 (0.705–0.720) | 0.998 (0.998–0.998) | 0.999 (0.999–0.999) |
| | MDCR | 0.716 (0.707–0.725) | 0.597 (0.588–0.606) | 0.997 (0.997–0.997) | 0.998 (0.998–0.998) |
| | Optum EHR | 0.641 (0.628–0.653) | 0.742 (0.730–0.755) | 0.999 (0.999–0.999) | 0.999 (0.999–0.999) |

*(Continued)*

**Table 4.** (Continued)

| Phenotype Algorithm | Database | Sensitivity (95% CI) | PPV (95% CI) | Specificity (95% CI) | NPV (95% CI) |
|---|---|---|---|---|---|
| Systemic lupus erythematosus incident with 2nd diagnosis code and correction for index date | CCAE | 0.209 (0.198–0.219) | 0.950 (0.937–0.961) | 1.000 (1.000–1.000) | 0.998 (0.997–0.998) |
| | Optum | 0.205 (0.197–0.214) | 0.922 (0.909–0.933) | 1.000 (1.000–1.000) | 0.996 (0.996–0.996) |
| | Germany | 0.264 (0.210–0.325) | 0.771 (0.666–0.856) | 1.000 (1.000–1.000) | 1.000 (1.000–1.000) |
| | JMDC | 0.355 (0.328–0.383) | 0.563 (0.527–0.598) | 1.000 (1.000–1.000) | 1.000 (1.000–1.000) |
| | MDCD | 0.234 (0.227–0.242) | 0.942 (0.933–0.950) | 1.000 (1.000–1.000) | 0.995 (0.995–0.995) |
| | MDCR | 0.175 (0.168–0.183) | 0.924 (0.911–0.935) | 1.000 (1.000–1.000) | 0.995 (0.995–0.995) |
| | Optum EHR | 0.251 (0.240–0.263) | 0.994 (0.988–0.997) | 1.000 (1.000–1.000) | 0.998 (0.998–0.998) |
| Systemic lupus erythematosus prevalent with 2nd diagnosis code and correction for index date | CCAE | 0.582 (0.570–0.594) | 0.961 (0.954–0.967) | 1.000 (1.000–1.000) | 0.999 (0.999–0.999) |
| | Optum | 0.644 (0.634–0.654) | 0.943 (0.937–0.948) | 1.000 (1.000–1.000) | 0.998 (0.998–0.998) |
| | Germany | 0.636 (0.572–0.697) | 0.782 (0.717–0.837) | 1.000 (1.000–1.000) | 1.000 (1.000–1.000) |
| | JMDC | 0.559 (0.530–0.587) | 0.576 (0.547–0.604) | 1.000 (1.000–1.000) | 1.000 (1.000–1.000) |
| | MDCD | 0.582 (0.573–0.590) | 0.935 (0.929–0.941) | 1.000 (1.000–1.000) | 0.997 (0.997–0.997) |
| | MDCR | 0.468 (0.458–0.478) | 0.946 (0.939–0.952) | 1.000 (1.000–1.000) | 0.997 (0.997–0.997) |
| | Optum EHR | 0.347 (0.335–0.360) | 0.994 (0.989–0.997) | 1.000 (1.000–1.000) | 0.998 (0.998–0.998) |

CI–Confidence interval; PPV–Positive predictive value; NPV–Negative predictive value; CCAE—IBM® MarketScan® Commercial Database; Optum—Optum's Clinformatics® Data Mart; Germany—IQVIA Disease Analyzer–Germany; JMDC—Japan Medical Data Center; MDCD IBM® MarketScan® Multi-State Medicaid Database; MDCR—IBM® MarketScan® Medicare Supplemental Database; Optum EHR—Optum's longitudinal EHR repository; DM–Dermatomyositis; SSc Systemic sclerosis.

## Discussion

In this study we used a data-driven approach to develop phenotype algorithms for SLE. The final four selected algorithms were developed to discriminate between incident and prevalent SLE populations, each with either higher sensitivity or higher specificity. In these algorithms, corrections were applied to address possible index date misclassification that might occur if the algorithms were limited to only diagnosis codes for SLE by adjusting the index date accounting for prior drugs, signs and symptoms. We validated each algorithm using the Phe-Valuator method to estimate the sensitivity, specificity, and positive and negative predictive values. To our knowledge, these are the first empirically derived phenotype algorithm definitions for SLE with the full set of validation performance characteristics. These performance characteristics may be used in studies requiring quantitative bias analysis (QBA). QBA provides quantitative estimates of the direction, magnitude, and uncertainty arising from systematic errors [16]. QBA, for example, may be used to estimate systematic errors in incidence rates, providing a more robust estimate. In QBA, the sensitivity and the specificity of the phenotype algorithm used within the study are included as parameters in the corrective equations.

The results from the current study may be used for these corrections. The use of these algorithms with QBA in SLE observational research may improve the confidence in study results.

Researchers may choose an algorithm for use in their observational studies based on fit for function according to the estimated performance characteristics. We found a large increase in PPV when a second code for SLE was added to the algorithm with a concomitant large decrease in sensitivity compared to the single code algorithm. These differences indicate that both the single code and the two code algorithms should be considered for use in studies depending on the study requirements. For studies requiring a more sensitive algorithm, the Prevalent, 1X algorithm should be used; for studies requiring a more specific algorithm, the Incident or Prevalent, 2X algorithm should be used.

There are several strengths to the present study. First, this study developed phenotypes using data from nine large datasets covering five countries and reflect subjects of a wide range of ages and from various socioeconomic backgrounds. Many previous studies examining the performance characteristics of algorithms for SLE used smaller datasets [17–20]. The data used for the development of the phenotypes was analyzed using multiple approaches, providing ancillary verification for each of our decisions in determining algorithm logic. The approach we used in this study uses publicly available, open-source software providing the capability for full result replication. Included in the supplemental information are the JSON files which provide fully reproducible phenotype algorithms. There were also several limitations to our study, which included the use of administrative datasets primarily maintained for insurance billing that is well-known to have significant deficits including coding inaccuracies [21]. In addition, the estimation of performance characteristics using the PheValuator methodology is dependent on the quality of the data in the dataset, which can vary substantially [22]. Incomplete signs and symptoms documentation in data could affect the accuracy of the index date. This may reflect differences between insurance claims databases, e.g., CCAE and EHR databases, e.g, Optum EHR. The generalizability of findings to uninsured populations may differ from the insured population observed in this study. Prescription drug treatments in claims and EHR data are not specifically associated with an indication which could affect the signs and symptoms and correction for index date misclassification and associated metrics. The algorithm validation was performed using a method involving predictive modeling of SLE rather than case review. This method does have the advantage of providing performance characteristics for multiple databases. It also provides the full set of performance metrics including sensitivity and specificity which are rarely provides in validation studies using case review [10]. Using this method we found similar results to a prior validation of SLE. The results from PheValuator have also been compared to the results from previously published validation studies and have demonstrated excellent agreement [12]. In the algorithm where an incident cohort definition was defined with only one diagnostic code it was not possible to determine if any of these were rule out diagnoses. Lastly, our study observed data from those subjects who presented for medical attention; those who did not seek medical attention but had the disease were not included and may affect the metrics in this study. Those with less severe disease may also have not sought medical attention.

## Conclusions

This study developed and thoroughly evaluated phenotype algorithms for SLE based on a data-driven approach. The results of this effort yielded four final algorithms, which may be applied by other researchers to observational studies of SLE. The four algorithms include options for prevalent vs. incident cohorts as well as sensitive vs. specific definitions. The validation metrics provided with these algorithms increases the confidence in these algorithms to identify SLE subjects and also enables the application of quantitative bias analysis.

## Supporting information

**S1 File. Description of and results from the literature search.**
(DOCX)

**S2 File. Description of the process for phenotype algorithm development.**
(DOCX)

**S3 File. The complete set of ICD, Read, and SNOMED codes used in the algorithms.**
(XLSX)

## Acknowledgments

The authors would like to thank Gayle Murry for her expert work on the literature search.

## Author Contributions

**Conceptualization:** Joel N. Swerdel, Darmendra Ramcharran.

**Data curation:** Joel N. Swerdel.

**Formal analysis:** Joel N. Swerdel, Jill Hardin.

**Investigation:** Joel N. Swerdel.

**Methodology:** Joel N. Swerdel, Jill Hardin.

**Project administration:** Joel N. Swerdel.

**Software:** Joel N. Swerdel.

**Supervision:** Joel N. Swerdel.

**Visualization:** Joel N. Swerdel.

**Writing – original draft:** Joel N. Swerdel.

**Writing – review & editing:** Darmendra Ramcharran, Jill Hardin.

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
