## [Decision Letter · Decision Letter 0]

14 Aug 2022

PONE-D-22-09946Using a Data-driven Approach for the Development and Evaluation of Phenotype Algorithms for Systemic Lupus ErythematosusPLOS ONE

Dear Dr. Swerdel,

Thank you for submitting your manuscript to PLOS ONE. After careful consideration, we feel that it has merit but does not fully meet PLOS ONE’s publication criteria as it currently stands. Therefore, we invite you to submit a revised version of the manuscript that addresses the points raised during the review process.

The manuscript has been assessed by two reviewers, and their comments are appended below. The reviewers have raised concerns about the level of detail provided in the methods section, and the clarity of some aspects of the developmental process and cohort selection. Additionally, it was noted that limitations are not adequately discussed. 

Could you please revise the manuscript to carefully address the concerns raised?

We look forward to receiving your revised manuscript.

Kind regards,

Clare Mc Fadden

Editorial Office

PLOS ONE

Journal Requirements:

Authors JS, DR, and JH are employees of Janssen Research and Development and shareholders of Johnson & Johnson.

Reviewers' comments:

Reviewer's Responses to Questions

**Comments to the Author**

1. Is the manuscript technically sound, and do the data support the conclusions?

Reviewer #1: Partly

Reviewer #2: Partly

2. Has the statistical analysis been performed appropriately and rigorously? 

Reviewer #1: Yes

Reviewer #2: Yes

3. Have the authors made all data underlying the findings in their manuscript fully available?

Reviewer #1: Yes

Reviewer #2: No

4. Is the manuscript presented in an intelligible fashion and written in standard English?

Reviewer #1: Yes

Reviewer #2: Yes

5. Review Comments to the Author

Reviewer #1: In this study, the authors use 9 different administrative data sets from around the world to develop phenotype algorithms for systemic lupus erythematosus. The study has several strengths. First, it uses multiple data sets and larger numbers of patients than prior studies. Second, there is a correction for misclassification of the index date by allowing for signs and symptoms of lupus to mark the index date if within 90 days of the first lupus diagnosis code. The source code is made available to increase reproducibility. Finally, they use a predictive algorithm previously published (PheValuator) as the gold standard for likelihood of the presence of lupus.

However, there are several weaknesses that must be addressed. There is no nested subset in which clinical diagnosis of SLE is compared against the Phevaluator likelihoods. Thus, there is no true sensitivity, specificity, and positive predictive value without the clinical goal standard. This should be discussed more in the methods as well as a limitation in the discussion.

The paper is very dense and sometimes difficult to read about rereading.

In table 1 and figure 2, 1 should use the same nomenclature as in the methods (i.e. incident 1x. etc. instead of cohort 1, 2, 3…). In figure 2, it is very difficult to tell what this is trying to depict the way it is currently labeled. It might be easier to understand if the Legend distinguishing blue and gray would simply say (gray = incident plus prevalent, blue equals incident). The top can then be labeled single diagnosis code, (1x) while the bottom can be labeled two diagnosis codes (2x).

It is difficult to understand how a minimum look back of 365 days was required for incident SLE, when not all datasets contain data spanning 365 days (IQVIA, table 2).

It is very concerning that low back pain was considered an early manifestation of lupus, as this is not a manifestation of lupus at all. How does removing this affect the correction for misclassified index date?

On page 13 the concept of a 3X algorithm is introduced in table 4, but it is not described in the methods for in the result text.

On page 13, fifth line from the bottom of the sentence "this is decreased to 39%…” Does not describe the condition under which the sensitivity reduced versus the preceding sentence. This is confusing.

In the discussion, a bit more explanation of QBA may be necessary.

Reviewer #2: The authors present a manuscript documenting the estimated performance of a new phenotyping algorithm for Systemic Lupus Erythematosus (SLE). This study uses a previously published tool by the lead author. The strengths of this paper include the number of datasets, published algorithm implementation for comparison, and the additional resources provided. However, the evidence provided for the performance of the algorithm is weak and it is unclear how the approach is “data-driven” as per the title and conclusion. This this manuscript suggests a large body of work that contributes to study of an important complex disease, but it read to this reviewer as an extended technical demonstration.

General Comments:

1) On the development of the algorithm presented: While the authors refer to a large amount of work behind their development process, there is little transparency into the factors that ultimately defined their decision making. This is also true of the “data-driven” and “empirical evidence” phrases used for “development and evaluation”; it is not clear how it was applied to the development portion. For example, the authors describe a literature search identifying an impressive 59 articles! The details of this investigation are not included, and the treatment of these findings amounts to selecting the condition codes and signs/symptom/treatment codes from several studies. What was used to establish the 90-day look-back window as “optimal” is not discussed, either. The authors may be doing everything according to best practices, but it is not shared with the audience. The start of the discussion reads as “The final four selected algorithms”, which again implies there was much more happening behind the scenes for this selection.

2) On the validation of the algorithms using PheValuator: Acknowledging the Dr. Swerdel is the first author on both this and the Phevaluator manuscript, please allow a brief summary: PheValuator is a tool that allows estimation of algorithm performance characteristics by generating a fuzzy silver standard on a population for evaluation by taking a strong definition of cases and controls and using a classification method to estimate likelihood across the entire population. The reliability of the estimates of algorithm performance are strongly predicated on the reliability of the ‘extremely specific (“xSpec”), sensitive, and prevalence cohorts’. Biases in those cohorts or unaccounted for differences in the source data can greatly impact the reliability of the resulting estimates. However, the authors do not make clear or justify the details of this cohort selection, nor provide information readers might use to evaluate the reliability of these estimates. In attempting to find the details, it looks like this OHDSI Atlas cohort definition describes it: https://github.com/OHDSI/PhenotypeEvaluations/blob/main/SLE/inst/cohorts/22370.json . I struggled to interpret exactly what this means, but it looks like it is expecting only a couple of SLE codes with 1 year of observation and perhaps a 21-day window separating codes. The accuracy of this interpretation aside, it should be clear to the reader what was done and justified as a foundation for the downstream estimation.

3) The authors describe several features of the identified cohorts and in comparison with the general population. These descriptions may benefit from a clinical perspective. For example on page 12: “A higher proportion of females compared to males with SLE were identified. The largest disproportionality was in MDCD where 91% of the subjects were female.” This seems expected for this population. Perhaps making more of these details available broadly while noting methodologically or clinically significant details in the discussion would be helpful to readers. Grounding this work in the user’s/reader’s needs and helping address questions of “can I use this with my data” or “are there biases I may wish to further address by extending this approach” may be very powerful.

4) The names and references among the materials are not always consistent, which can make it hard to understand and connect the data provided. The SLE Cohort Diagnostics tool is neat to see, but the Cohort numbers in the tool do not align with the Cohort numbers in Figure 2, nor do the Cohort IDs in the tool match the Cohort IDs in the Github Repository. Table 2 is not sorted the same as Table 3 (eg, by name and not by abbreviation).

5) There are some areas where clarity could be improved with regards to the cohort selection and process which is especially important when the “any non-case is a control”. For example, it wasn’t clear if the incident population evaluation only considered individuals with at least 365 days of data (as they could not possibly qualify as a case).

Specific comments:

In the Cohort Diagnostics report, some of the cohorts appear to have extraneous information, eg the exclusion condition concept set in: https://github.com/OHDSI/PhenotypeEvaluations/blob/main/SLE/inst/cohorts/22370.json

Table 3: Including denominators for these (or presenting additionally as rates) would be helpful for interpretation.

Page 5, line 100: This appendix appears to only contain the SNOMED codes, not the expanded / mapped set of codes as described in the text.

Page 10, “ex-US databases”. Is this intended to be “non-US” or “extra-US”?

Page 11, text lines 7-8: The differences in this and the definitions of the algorithms, ie, these statistics consider a window based on the index date while the algorithm window is defined based on the SLE code date, seem to make the numbers harder to interpret on the surface.

Page 12, “MDCD where about 25% of the first diagnoses were made in an emergency room visit”- It may be worth interpreting this later. Is this a feature of the population that is expected or is it a signal that the algorithm is not performing as expected or something else?

Page 12, near end: Particularly given the international cohorts used, specifying the Clinical Modification versions of the ICD systems were applicable is important for clarity.

Page 13, “Rates in Australia and France varied considerably, likely due to the small sample size.” And “Due to low subject counts, we were unable to calculate the performance characteristics for Australia and France.”. This merits further treatment. If the authors method cannot analyze 4 or 5 million person datasets, that suggests challenges for many potential users who do not have access to such large sets. Is this a matter of the low prevalence of SLE? Was there a metric that showed you could not use datasets of this size, or some error reported by PheValuator? It would be helpful for readers and users/implementers to understand. The data presented suggests to this reviewer that perhaps the algorithm does not work in these datasets- comparing the IQVIA France and Germany datasets there are dramatically different rates observed (considering Tables 2 and 3). One might conjecture that the IQVIA GP data is insufficient for SLE identification as it did function in Germany but neither France nor Australia, though there are certainly other possibilities.

6. PLOS authors have the option to publish the peer review history of their article (what does this mean?). If published, this will include your full peer review and any attached files.

Reviewer #1: No

Reviewer #2: No

---

## [Author Response · Author response to Decision Letter 0]

26 Sep 2022

Reviewer Response

Reviewer #1: In this study, the authors use 9 different administrative data sets from around the world to develop phenotype algorithms for systemic lupus erythematosus. The study has several strengths. First, it uses multiple data sets and larger numbers of patients than prior studies. Second, there is a correction for misclassification of the index date by allowing for signs and symptoms of lupus to mark the index date if within 90 days of the first lupus diagnosis code. The source code is made available to increase reproducibility. Finally, they use a predictive algorithm previously published (PheValuator) as the gold standard for likelihood of the presence of lupus.

However, there are several weaknesses that must be addressed. There is no nested subset in which clinical diagnosis of SLE is compared against the Phevaluator likelihoods. Thus, there is no true sensitivity, specificity, and positive predictive value without the clinical goal standard. This should be discussed more in the methods as well as a limitation in the discussion.

Thank you for bringing up this very important point. While the validation method, PheValuator, does not use chart review for validation, we think the predicted probabilities accurately assess the likelihood of SLE in our trained cohort. This thinking is strengthened by our recent publication (https://www.sciencedirect.com/science/article/pii/S1532046422001885) where we compared the results from PheValuator to previously published chart validation studies for 17 different phenotypes encompassing 86 phenotype algorithms. We found close agreement between PheValuator results and those from chart review. We have included the following in the methods section:

Page 9, Line 11:

“While algorithm validation results from chart review are considered the “gold standard”, we have compared the results from PheValuator with prior studies using chart review and found excellent agreement between the two methods. (12)”

In the limitations section:

Page 18, Line 45:

“The results from PheValuator have also been compared to the results from previously published validation studies and have demonstrated excellent agreement.(12)”

The paper is very dense and sometimes difficult to read about rereading.

In table 1 and figure 2, 1 should use the same nomenclature as in the methods (i.e. incident 1x. etc. instead of cohort 1, 2, 3…). 

Thank you for your comment. This has been corrected in both Table 1 and Figure 2.

In figure 2, it is very difficult to tell what this is trying to depict the way it is currently labeled. It might be easier to understand if the Legend distinguishing blue and gray would simply say (gray = incident plus prevalent, blue equals incident). The top can then be labeled single diagnosis code, (1x) while the bottom can be labeled two diagnosis codes (2x).

Thank you for this correction. We have made this change in Figure 2.

It is difficult to understand how a minimum look back of 365 days was required for incident SLE, when not all datasets contain data spanning 365 days (IQVIA, table 2).

Thank you for this observation. While the median follow-up time in some of the databases was indeed less than one year, we were able to observe patients with 365 day lookback as shown in Table 3. The lookback time requirement did have an impact on the number of subjects observed. For example, in IQVIA France the number of subjects in the incident cohort requiring 365 day lookback was 117 whereas the number of subjects in the prevalent cohort was 260, over a 50% reduction, obviously impacting the sensitivity of the incident algorithm.

We have added the following to Page 5, Line 88:

“Requiring a 365 day look-back for the incident algorithm will impact the sensitivity of the algorithm especially in databases where subjects are observed for shorter periods of time.”

It is very concerning that low back pain was considered an early manifestation of lupus, as this is not a manifestation of lupus at all. How does removing this affect the correction for misclassified index date?

Thank you for this comment. The high prevalence of back pain in SLE was found by Cezarino et al (2017). We have added this citation to the list of citations on Page 5, Line 96. The method we used to determine possible index date misclassification is empirically based and we included low back pain based on the relatively high prevalence of low back pain in the SLE population (> 15%) prior to initial diagnosis of SLE. Our prevalence of low back pain was similar to Cezarino et al.

On page 13 the concept of a 3X algorithm is introduced in table 4, but it is not described in the methods for in the result text.

Thank you for this correction. We have added the following line to Page 9, Line 7:

“The two algorithms from Barnado et al were “Systemic Lupus 3X plus ever anti-malarial drugs” and “Systemic Lupus 3X plus ever anti-malarial drugs excluding dermatomyositis and systemic sclerosis”.”

On page 13, fifth line from the bottom of the sentence "this is decreased to 39%…” Does not describe the condition under which the sensitivity reduced versus the preceding sentence. This is confusing.

Thank you for picking up this error. The comparison was to the incident cohorts and the sentence has been corrected at Page 14, Line 108 from:

“This decreased to 39% (single code required) and 24% (two codes required).”

To: 

“This decreased to 39% (single code required) and 24% (two codes required) for the two incident cohorts.”

In the discussion, a bit more explanation of QBA may be necessary.

We appreciate your comment. We have added the following to Page 17, Line 13:

“In QBA, the sensitivity and the specificity of the phenotype algorithm used within the study are included as parameters in the corrective equations. The results from the current study may be used for these corrections.”

Reviewer #2: The authors present a manuscript documenting the estimated performance of a new phenotyping algorithm for Systemic Lupus Erythematosus (SLE). This study uses a previously published tool by the lead author. The strengths of this paper include the number of datasets, published algorithm implementation for comparison, and the additional resources provided. However, the evidence provided for the performance of the algorithm is weak and it is unclear how the approach is “data-driven” as per the title and conclusion. This this manuscript suggests a large body of work that contributes to study of an important complex disease, but it read to this reviewer as an extended technical demonstration.

General Comments:

1) On the development of the algorithm presented: While the authors refer to a large amount of work behind their development process, there is little transparency into the factors that ultimately defined their decision making. This is also true of the “data-driven” and “empirical evidence” phrases used for “development and evaluation”; it is not clear how it was applied to the development portion. For example, the authors describe a literature search identifying an impressive 59 articles! The details of this investigation are not included, and the treatment of these findings amounts to selecting the condition codes and signs/symptom/treatment codes from several studies. 

The goal of this paper was to provide researcher with empirically-derived and validated phenotype algorithms that they may use in their research on SLE. We think that the methodology that we used to derive and validate these algorithms are a significant improvement over previous studies that published algorithms derived with less evidence. The rigor applied in our algorithm development process means that the algorithms may be used directly within other’s research studies with confidence of reduced bias. We respectfully disagree with the assessment that the evidence provided for the performance of the algorithm is weak. As noted in comment to Reviewer 1, we have now shown in our latest publication on PheValuator (https://pubmed.ncbi.nlm.nih.gov/35995107/) that the results from this tool compare favorably with prior validation studies using chart validation.

Your comments about our methods for this empirically- driven process being unclear are well received. In order to conserve space in the article we provided a very brief description of the process. This was not sufficient. To remedy that error, we have provided two appendices to be included in the on-line supplemental information. Supplemental information 1 provides details of the literature search including the search strategy in PUBMED, the authors and citations for the articles we found, and a table of the diagnosis codes used within algorithms in the previously published articles. Supplemental information 2 provides a flow diagram and description of the complete process we used to develop the algorithms using our data-driven approach. 

To provide better clarity for the process we used to develop the algorithms we have changed the text on Page 4, Line 79 from:

“After reviewing the results from the literature search and determining all the diagnosis codes for SLE in the different vocabularies, i.e., International Classification of Diseases, Ninth (ICD-9) or Tenth (ICD-10) Revision and Read codes, codes were translated each into the Systemized Nomenclature of Medicine (SNOMED) vocabulary using the OHDSI open-source ATLAS tool (https://github.com/OHDSI/Atlas). Translating codes from disparate vocabularies into SNOMED has been shown to be effective and improves the efficiency and transportability of research.(7)”

To:

“We applied a rigorous process to develop the algorithms in this study. We used an extensive literature review (details and results are in Supplemental information 1), along with several tools within the Observational Health Data Sciences and Informatics (OHDSI) tool stack for empirical analysis. The goal of instituting this process was to enhance the science of phenotype algorithm development. The full details of the process are detailed in Supplemental information 2.”

What was used to establish the 90-day look-back window as “optimal” is not discussed, either. 

Thank you for this observation. We chose a 90-day look back window as optimal as the rates for the signs and symptoms for SLE were highest during this period and significantly reduced in prevalence prior to 90 days. We have added to Page 5, Line 93:

“In the period 365 to 91 days prior to the first diagnosis code for SLE, the rates of these signs and symptoms were significantly lower than in the 90 to 0 days prior to index date. Thus, the use of 90 days prior to index as the optimal time period choice.”

The authors may be doing everything according to best practices, but it is not shared with the audience. The start of the discussion reads as “The final four selected algorithms”, which again implies there was much more happening behind the scenes for this selection.

Thank you for pointing out this lack of detail. We have increased the level of detail as described above using two new documents as supplemental information .

2) On the validation of the algorithms using PheValuator: Acknowledging the Dr. Swerdel is the first author on both this and the PheValuator manuscript, please allow a brief summary: PheValuator is a tool that allows estimation of algorithm performance characteristics by generating a fuzzy silver standard on a population for evaluation by taking a strong definition of cases and controls and using a classification method to estimate likelihood across the entire population. The reliability of the estimates of algorithm performance are strongly predicated on the reliability of the ‘extremely specific (“xSpec”), sensitive, and prevalence cohorts’. Biases in those cohorts or unaccounted for differences in the source data can greatly impact the reliability of the resulting estimates. However, the authors do not make clear or justify the details of this cohort selection, nor provide information readers might use to evaluate the reliability of these estimates. 

Thank you for your comment. We felt understanding the full details of the PheValuator process is beyond the scope of this article. We use many open-source tools within our process the details and source code of which are available on-line. As Reviewer 1 noted the article is dense in detail, this agrees with our conclusion to not include more detail on the utilized tools but relies on the interested reader to use the on-line references for more detail. Also, based on a comment from Reviewer 1, we have added additional detail on the validity of the PheValuator process in the Methods section. The results presented in the PheValuator follow-up article demonstrate the close agreement of the results from PheValuator with the results from previously published articles on phenotype algorithm validation using chart review. We feel this agreement justifies the structure of the cohorts in the PheValuator process. Also, as noted above, we have provided significantly more detail on the overall process including the PheValuator process in supplemental information documents.

Page 9, Line 11:

“While algorithm validation results from chart review are considered the “gold standard”, we have compared the results from PheValuator with prior studies using chart review and found excellent agreement between the two methods. (12)”

In attempting to find the details, it looks like this OHDSI Atlas cohort definition describes it: https://github.com/OHDSI/PhenotypeEvaluations/blob/main/SLE/inst/cohorts/22370.json . I struggled to interpret exactly what this means, but it looks like it is expecting only a couple of SLE codes with 1 year of observation and perhaps a 21-day window separating codes. The accuracy of this interpretation aside, it should be clear to the reader what was done and justified as a foundation for the downstream estimation.

This is an important comment. The details of the PheValuator process used for the analyses within this paper are thoroughly described in the article published since the SLE article was submitted for review. The details of the process for interested readers are cited above (and now in the SLE article) and may be found at https://pubmed.ncbi.nlm.nih.gov/35995107/. We have also provided significantly more detail on the overall process including the PheValuator process in supplemental information documents.

3) The authors describe several features of the identified cohorts and in comparison with the general population. These descriptions may benefit from a clinical perspective. For example on page 12: “A higher proportion of females compared to males with SLE were identified. The largest disproportionality was in MDCD where 91% of the subjects were female.” This seems expected for this population. Perhaps making more of these details available broadly while noting methodologically or clinically significant details in the discussion would be helpful to readers. Grounding this work in the user’s/reader’s needs and helping address questions of “can I use this with my data” or “are there biases I may wish to further address by extending this approach” may be very powerful.

Thank you for the suggestion. The goal of this paper was to provide researchers empirically-derived and validated phenotype algorithms for studies of SLE. We feel we additionally provided a wealth of information that readers may use for their own hypothesis generation based upon the data within the cohort diagnostics shiny application. We did provide some insights into the data as you noted above. However, a thorough clinical investigation of the data is beyond the scope of this research article. We felt that a future separate article describing the clinical ramifications of the SLE phenotype algorithms was more appropriate.

4) The names and references among the materials are not always consistent, which can make it hard to understand and connect the data provided. The SLE Cohort Diagnostics tool is neat to see, but the Cohort numbers in the tool do not align with the Cohort numbers in Figure 2, nor do the Cohort IDs in the tool match the Cohort IDs in the Github Repository. Table 2 is not sorted the same as Table 3 (eg, by name and not by abbreviation).

Thank you for noting these errors. We have re-sorted Tables 2 and 3 and now they are in the same sort order based on abbreviation. We have renamed the .json files in the Github repository to match the names in Table 2 and eliminated duplicate cohorts in the repository. We have also added a reference column in Table 2 to show the cohort numbers in the cohort diagnostic shiny application.

5) There are some areas where clarity could be improved with regards to the cohort selection and process which is especially important when the “any non-case is a control”. For example, it wasn’t clear if the incident population evaluation only considered individuals with at least 365 days of data (as they could not possibly qualify as a case).

Thank you for this comment. In the analyses from PheValuator, cohorts are matched to possible controls based on the required observation period of the algorithm. For example, in the incident algorithms requiring 365-day lookback, possible controls are from subjects where their compared time period had at least 365 days prior lookback.

Specific comments:

In the Cohort Diagnostics report, some of the cohorts appear to have extraneous information, eg the exclusion condition concept set in: https://github.com/OHDSI/PhenotypeEvaluations/blob/main/SLE/inst/cohorts/22370.json

Thank you for your diligent effort in finding that error. All the cohorts have been re-checked and all extraneous concept sets have been removed.

Table 3: Including denominators for these (or presenting additionally as rates) would be helpful for interpretation.

Thank you for the comment. The denominators for these counts are the entire population for each database. The database population may be found in the sixth column of Table 2 (Number of Persons (millions)). The incidence rates can be found for different strata (e.g., age group, sex) in the shiny app, for example:

Page 5, line 100: This appendix appears to only contain the SNOMED codes, not the expanded / mapped set of codes as described in the text.

Thank you for finding this error. The new table in supplemental information 3 now contains all the SNOMED codes as well as the codes from the native classification schemes, e.g., ICD-9.

Page 10, “ex-US databases”. Is this intended to be “non-US” or “extra-US”?

Thank you for pointing out this confusing usage. We have made the change from “ex-US databases” to “databases from outside the US” on Page 10, line 33.

Page 11, text lines 7-8: The differences in this and the definitions of the algorithms, ie, these statistics consider a window based on the index date while the algorithm window is defined based on the SLE code date, seem to make the numbers harder to interpret on the surface.

The final analyses for differences in the algorithms are all compared between algorithms with the index date correction applied. The analyses use the same windows from the index date and no longer use windows starting from the SLE code date only (the original index date prior to correction). For example, in the line you have cited, the differences were between the 2X algorithm with the index date corrected and the 1X algorithm with the index date corrected. As we consider the index date correction changes the index date to a date closer to the start of the clinical occurrence of SLE, these comparisons are now appropriate.

Page 12, “MDCD where about 25% of the first diagnoses were made in an emergency room visit”- It may be worth interpreting this later. Is this a feature of the population that is expected or is it a signal that the algorithm is not performing as expected or something else?

Excellent insight, thank you. It is well known that Medicaid recipients use the emergency room more than other health insurance subscribers. To address the issue we have added the following to Page 12, Line 73:

“This follows with other published studies examining the higher use of the emergency room in MDCD recipients.(15)”

Page 12, near end: Particularly given the international cohorts used, specifying the Clinical Modification versions of the ICD systems were applicable is important for clarity.

We have changed the text at Page 12, Line 80 from:

“In the Incident, 2X algorithm, the most prevalent index event was a diagnosis code of “Systemic lupus erythematosus” (SNOMED code 257628; ICD-10 M32.9 (“Systemic lupus erythematosus, unspecified”); ICD-9 710.0).”

To:

“In the Incident, 2X algorithm, the most prevalent index event was a diagnosis code of “Systemic lupus erythematosus” (SNOMED code 257628; ICD-10CM and ICD-10GM M32.9 (“Systemic lupus erythematosus, unspecified”); ICD-9CM 710.0 (“Systemic lupus erythematosus”)).”

Page 13, “Rates in Australia and France varied considerably, likely due to the small sample size.” And “Due to low subject counts, we were unable to calculate the performance characteristics for Australia and France.”. This merits further treatment. If the authors method cannot analyze 4 or 5 million person datasets, that suggests challenges for many potential users who do not have access to such large sets. Is this a matter of the low prevalence of SLE? Was there a metric that showed you could not use datasets of this size, or some error reported by PheValuator? It would be helpful for readers and users/implementers to understand. The data presented suggests to this reviewer that perhaps the algorithm does not work in these datasets- comparing the IQVIA France and Germany datasets there are dramatically different rates observed (considering Tables 2 and 3). One might conjecture that the IQVIA GP data is insufficient for SLE identification as it did function in Germany but neither France nor Australia, though there are certainly other possibilities.

There are multiple factors likely contributing to the low subject counts in France and Australia. The likely reason is that these two databases, as well as the IQVIA German database, are exclusively general practitioner databases. It is likely that many subjects with SLE are treated by specialists. To aid the reader we are adding the following on Page 13, Line 91:

“The small number of SLE subjects in the Australian and French databases is likely due to the database being limited to general practitioners. In other databases which include specialists, such as rheumatologists, the sample size and incidence rates are higher and more stable.”

There are limitations to the use of PheValuator regarding sample size. The limiting factor is the xSpec cohort which requires at least 200 subjects for the development of an accurate model. This subject count was not satisfied in several of the databases. To clarify this point we are adding to Page 13, Line 100:

“PheValuator requires a minimum of 200 subjects with a high likelihood of having SLE to produce an accurate model. This number was not satisfied in the Australian or French databases. As noted earlier, this may be due to the limitation of these databases to general practitioners.”

---

## [Editor Report · Decision Letter 1]

5 Feb 2023

Using a Data-driven Approach for the Development and Evaluation of Phenotype Algorithms for Systemic Lupus Erythematosus

PONE-D-22-09946R1

Dear Dr. Swerdel,

We’re pleased to inform you that your manuscript has been judged scientifically suitable for publication and will be formally accepted for publication once it meets all outstanding technical requirements.

Kind regards,

Luca Navarini

Academic Editor

PLOS ONE

---

## [Editor Report · Acceptance letter]

8 Feb 2023

PONE-D-22-09946R1 

Using a Data-driven Approach for the Development and Evaluation of Phenotype Algorithms for Systemic Lupus Erythematosus 

Dear Dr. Swerdel:

I'm pleased to inform you that your manuscript has been deemed suitable for publication in PLOS ONE. Congratulations! Your manuscript is now with our production department. 

Kind regards, 

on behalf of

Dr. Luca Navarini 

Academic Editor

PLOS ONE